# Identification of Novel lncRNAs Differentially Expressed in Placentas of Chinese Ningqiang Pony and Yili Horse Breeds

**DOI:** 10.3390/ani10010119

**Published:** 2020-01-11

**Authors:** Yabin Pu, Yanli Zhang, Tian Zhang, Jianlin Han, Yuehui Ma, Xuexue Liu

**Affiliations:** 1Institute of Animal Science, Chinese Academy of Agricultural Sciences (CAAS), Beijing 100193, China; puyabin@caas.cn (Y.P.); Jenny_CAAS@163.com (Y.Z.); 2CAAS-ILRI Joint Laboratory on Livestock and Forage Genetic Resources, Institute of Animal Science, Chinese Academy of Agricultural Sciences (CAAS), Beijing 100193, China; h.jianlin@cgiar.org; 3State Key Laboratory of Dao-di Herbs, National Resource Center for Chinese Materia Medica, China Academy of Chinese Medical Sciences, Beijing 100700, China; lffmsfe@126.com; 4International Livestock Research Institute (ILRI), Nairobi 00100, Kenya

**Keywords:** pony, horse, long non-coding RNA, placenta, *TBX3*

## Abstract

**Simple Summary:**

The Ningqiang pony (NQ) is a famous breed with an average adult height at the withers (the ridge between horse’s shoulder blades) of less than 106 cm, but the genetic mechanism responsible for its small stature remains unclear. Research on Chinese pony breeds is a topic of theoretical and practical importance. Long non-coding RNA (lncRNA) has been shown to regulate many biological functions, including growth. Our study, for the first time, reports the lncRNAs transcribed in horse placentas. By comparing 3011 transcripts coding 1464 lncRNAs from NQ ponies and Yili horses (YL) with distinct body sizes, six (*TBX3* (*T-box3*), *CACNA1F* (*L-type voltage-dependent calcium channel a1F*), *EDN3* (*endothelin-3*), *KAT5* (*histone acetyltransferase KAT5*), *ZNF281* (*zinc finger protein 281*), *TMED2* (*transmembrane emp24 domain*), and *TGFB1* (*transforming growth factor beta 1*)) out of the 233 genes targeted by differentially expressed lncRNAs were identified as being involved in limb development, skeletal myoblast differentiation, and embryo development. These results provide new insights into the genetic architecture of horse body size.

**Abstract:**

As a nutrient sensor, the placenta plays a key role in regulating fetus growth and development. Long non-coding RNAs (lncRNAs) have been shown to regulate growth-related traits. However, the biological function of lncRNAs in horse placentas remains unclear. To compare the expression patterns of lncRNAs in the placentas of the Chinese Ningqiang (NQ) and Yili (YL) breeds, we performed a transcriptome analysis using RNA sequencing (RNA-seq) technology. NQ is a pony breed with an average adult height at the withers of less than 106 cm, whereas that of YL is around 148 cm. Based on 813 million high-quality reads and stringent quality control procedures, 3011 transcripts coding for 1464 placental lncRNAs were identified and mapped to the horse reference genome. We found 107 differentially expressed lncRNAs (DELs) between NQ and YL, including 68 up-regulated and 39 down-regulated DELs in YL. Six (*TBX3*, *CACNA1F*, *EDN3*, *KAT5*, *ZNF281*, *TMED2*, and *TGFB1*) out of the 233 genes targeted by DELs were identified as being involved in limb development, skeletal myoblast differentiation, and embryo development. Two DELs were predicted to target the *TBX3* gene, which was found to be under strong selection and associated with small body size in the Chinese Debao pony breed. This finding suggests the potential functional significance of placental lncRNAs in regulating horse body size.

## 1. Introduction

Non-coding RNAs (ncRNAs) are a heterogeneous class of RNA molecules transcribed from non(-protein)-coding regions in a genome because they lack an open reading frame and consequently have no protein-coding ability. The ncRNAs are classified into long ncRNAs (lncRNAs, >200 bp) and small ncRNAs (sncRNAs, 18–200 bp) according to their lengths [1,2]. lncRNAs play critical roles in many important biological processes, including cell proliferation and differentiation, signal transduction, stem cell maintenance and metabolism, genome imprinting, chromatin remodeling, regulation of cell cycle, and splicing regulation [3,4]. Many lncRNAs have been identified in a wide range of organisms [5,6], but have been poorly characterized in horses [7]. Only one single study has reported on the identification of horse lncRNAs [7]. It was based on eight types of tissues from 59 horses, producing over 20 million reads of RNA-sequencing (RNA-seq) data, building a database of 20,800 horse lncRNAs [7,8] and supporting the intensive identification of functional lncRNAs in horses.

As one of the five Chinese indigenous pony breeds (Debao, Jianchang, Guizhou, Yunnan, and Ningqiang), the Ningqiang pony (NQ) has an average adult height at withers of less than 106 cm [9]. It is distributed in the northeast of the Shaanxi province but is now close to being endangered. To adapt to the local mountainous environment, NQ has experienced a long history of both natural and human selection for its small stature and an extremely tough body conformation. The Yili horse (YL), a famous racing horse breed with an adult height at the withers of around 148 cm [9], was derived from the Kazakh horse through crossbreeding with Orlov Trotter, Don, Budyonny, and Akhal-Teke horse breeds in the 20th century [10]. The large variation in heights at the withers between NQ and YL provides an opportunity to study their underlying molecular mechanisms.

The placenta is essential for fetus growth and development as it acts as a nutrient sensor to regulate the transfer of nutrients from mother to fetus [11]. Studies have reported a strong association between neonatal body size and placental structure [12]. During pregnancy, the placenta is an important endocrine organ that produces numerous hormones, including estrogens and progesterone, growth hormone, insulin, and insulin-like growth factor 1, most of them crucial in the regulation of fetal growth [13]. A genomic scan showed that the *TBX3* gene is under strong positive selection in the Chinese Debao pony [14]; however, the functional validation of *TBX3* in horse tissues has not yet been performed due to lack of appropriate samples, i.e., placental tissue.

The development of whole-transcriptome sequencing technology provides an opportunity to efficiently identify and annotate horse lncRNAs. In this study, we conducted RNA-seq analysis of the placental tissues from NQ and YL. After stringent quality control and annotation, the differentially expressed placental lncRNAs between NQ and YL and their target genes were identified. Two lncRNAs were found to regulate *TBX3*, which may, therefore, play an important role in the development of horse body size. Our study facilitates the further exploration of the fundamental function of lncRNAs in the regulation of horse body size.

## 2. Materials and Methods

### 2.1. Ethics Statement

All procedures involving the handling of ponies and horses were approved by the Animal Care and Use Committee of the Chinese Academy of Agricultural Sciences and the Ministry of Agriculture of the People’s Republic of China (IASCAAS-AE-03).

### 2.2. Animals and Samples

The placental samples of NQ and YL were taken from the Ningqiang national conservation farm in Ningqiang county in Shaanxi province (log: 105°582′ and lat: 32°825′) and Zhaosu horse farm in Zhaosu county in Xinjiang province (log: 81.131 and lat: 43.155). Three YL mares with an average height of 141.3 cm and three NQ mares with an average height of 105 cm were taken. To ensure consistency between biological replications, adult mares about six years old were chosen for placental tissue sampling. To avoid relatedness between the collected samples, we consulted the local technicians and farmers regarding the pedigrees of the mares. These pregnant mares (multiparous) were randomly selected, ensuring no genetic kinship within three generations. After giving birth, the placental tissues from the mares were immediately collected by experienced veterinarians and stored in the RNAlater (Thermo Fisher Scientific, Waltham, MA, USA) at −80 °C. All samples were transported to our laboratory for RNA extraction.

### 2.3. RNA Extraction, Library Preparation, and Sequencing

Total RNA from all six placental samples was extracted by using the RNeasy Mini Kit (Qiagen, Hilden, Germany). The RNA was quantified according to the RNA integrity number (RIN) value >8 using an Agilent 2100 Bioanalyzer System (Agilent Technologies, Santa Clara, CA, USA). The Illumina TruSeq RNA Library Prep Kit v2 (Illumina, San Diego, CA, USA) was used to construct the cDNA library for RNA-seq. Briefly, after end-repair and adding poly (A) to the 3′ end of the RNA fragments, the sequencing linker was ligated and the products were purified for PCR amplification. The PCR products were separated by 2% agarose gel electrophoresis and fragments varying from 400 to 500 bp were selected to construct sequencing libraries. Finally, the six libraries were sequenced on an Illumina HiSeq 2500 System (Illumina, San Diego, CA, USA) using a 150 × 2 bp paired-end sequencing strategy by Beijing Compass Biotechnology Co. Ltd. (Beijing, China).

### 2.4. Trimming and Mapping of Reads

To remove low quality reads and adapters, raw sequencing reads were first filtered using Trimmomatic (version 0.33) software (http://www.usadellab.org/cms/index.php?page=trimmomatic) [15]. The reads were filtered using the following steps: first, reads with adapters were removed; second, reads with more than 10% ploy-N were discarded; and third, low-quality reads with more than 50% of bases and Phred scores less than 5 were removed. The Phred scores (Q20, Q30) and GC content of the clean data was calculated. All subsequent analyses were conducted based on the high-quality clean data. The quality of RNA-seq reads was ascertained using the FASTQC tool (http://www.bioinformatics.babraham.ac.uk/projects/fastqc). After indexing the reference assembly of horse genome EquCab3.0 with Bowtie-build software (https://sourceforge.net/projects/bowtie-bio/files/bowtie2) [16], all clean reads were mapped with the TopHat2 alignment tool (http://ccb.jhu.edu/software/tophat/index.shtml) [17] using default parameters. The mapped reads of each sample were assembled using Cufflinks v2.1.1 (http://cole-trapnell-lab.github.io/cufflinks/releases/v2.1.1) [18]. To identify the similarity between samples, the read counts were calculated using HTSeq (https://htseq.readthedocs.io/en/release_0.11.1) [19] and normalized using R package DESeq (https://bioconductor.org/packages/release/bioc/html/DESeq.html) [20].

### 2.5. Step-Wise Filtering for Transcripts

Cufflinks [18] (with the command of ‘min-frags-per-transfrag = 0 and -library-type’) was used to de novo assemble the transcripts from aligned RNA-seq reads and to obtain fragments per kilobase of transcript per million mapped fragments (FKPM) of each gene. Then, all the transcripts were merged using Cuffmerge software (http://cole-trapnell-lab.github.io/cufflinks/cuffmerge/index.html) [18].

### 2.6. Identification of lncRNAs from the Assembled Transcripts

lncRNAs were identified from the assembled transcripts following four steps: (1) Removal of lowly expressed transcripts with FPKM < 0.5, (2) removal of short transcripts < 200 bp and < 2 exons, (3) removal of the transcripts with protein-coding capability using the coding-non-coding index (CNCI) software (https://github.com/www-bioinfo-org/CNCI) with default parameters for a CNCI score ≥ 0 [21], and (4) removal of the transcripts mapped within the 1 kb flanking regions of an annotated gene. For incomplete UTR annotation of protein-coding genes in horses, the transcriptome data used in this study were based on the published refined transcriptome with the candidate lncRNA post-filter 3 removed [22]. The rest of the transcripts with more than 90% located to the “NM_” sequences were also removed. The pipeline used to identify putative lncRNAs from the RNA-seq data is presented in Appendix B
Figure A1.

### 2.7. Quantification of lncRNA Expression

Cuffdiff software (http://cole-trapnell-lab.github.io/cufflinks/cuffdiff/index.html) [18] was used to calculate differentially expressed lncRNAs (DELs) between YL and NQ breeds, with a corrected *p* ≤ 0.05 and fold change (FC) ≥ 2. Volcano plots of the DELs were employed to demonstrate the differences.

### 2.8. Prediction of Target Genes

Transcripts without coding potential composed our candidate set of lncRNAs. To explore the function of these potential lncRNAs, genes targeted by the placental lncRNAs were first predicted in *cis* action. The *cis* role refers to lncRNA action on neighboring target genes [23]. Coding genes from 100 kb upstream and downstream of the lncRNAs were searched. The *trans* role refers to the influence of lncRNA on the expressions of other genes. Pearson’s correlation coefficients (*r*) were calculated between the expression levels of the lncRNAs and mRNAs with customized scripts (*r* ≥ 0.95 or ≤−0.95). For the function of target genes, we referred to the relevant literature.

### 2.9. Gene Ontology (GO) Analysis

The functional annotation and pathway enrichment of the gene targeted by the DELs were performed using the online analysis tool Gene Ontology Consortium (http://www.geneontology.org/). This is a program combining the enriched GO terms and Kyoto Encyclopedia of Genes and Genomes (KEGG) pathways. The target genes were mapped to the GO database, and the hypergeometric test was used to determine which GO terms were significantly enriched among the target genes against the background of the reference horse genome EquCab3.0. GO terms were considered to be significantly enriched when they had corrected *p*-values less than 0.05. A similar method was used to determine which KEGG pathways and relevant complex biological terms were enriched among the target genes (threshold: corrected *p*-value < 0.05).

### 2.10. Quantitative Real-Time PCR (qRT-PCR) Validation

To further confirm the reliability of these identified lncRNAs, qRT-PCR assays were performed. First, the total RNA extracted from individual placentas of YL and NQ mares were used for validation. Then, the RNAs were inverse transcribed using the PrimeScript RT Reagent Kit with gDNA Eraser (TAKARA, RR047A, Beijing, China). The lncRNAs against the house-keeping gene *β-ACTN* (Beta-actin) as an endogenous control was quantified using the SYBR Green Master Mix (TAKARA, RR820A, Beijing, China). We randomly selected 10 lncRNAs for the quantification. The primers used in this study are listed in Table 1. qRT-PCR was performed using the following conditions: 95 °C for 60 s, then 40 cycles at 95 °C for 10 s, and the optimized annealing temperatures for 30 s in the Applied Biosystems^®^ 7500 Real-Time PCR System (Applied Biosystems, Foster City, CA, USA). Each reaction was performed in triplicate for each sample. Gene expression was quantified relative to *β-ACTN* expression using the comparative cycle threshold (ΔΔCT) method. Then, the log2Fold change (FC) of each gene expression level from qRT-PCR data was calculated for comparison with the expression patterns of RNA-seq data. Log2FC values were calculated for the comparison of YL with NQ.

## 3. Results

### 3.1. Overview of RNA-Seq Data

To identify the unique lncRNAs expressed in horse placenta, a prerequisite was integrating the high-quality and high-depth RNA-seq data. A total of 39.34–47.95 million raw reads were generated across the six samples. After discarding adaptor sequences, the median per-base quality was more than 30 for all the samples and the Q30 rate was more than 83.32% (Appendix A). Next, we mapped all the clean reads of every sample to the latest reference horse genome EquCab3.0 using TopHat2 software (http://ccb.jhu.edu/software/tophat/index.shtml) [17], achieving alignment rates of approximately 65.50–81.00%.

### 3.2. Identification of lncRNAs

Overall, 20,961 de novo transcripts were captured for analyses. To evaluate the quality of these transcripts, we used Cuffcompare software (http://cole-trapnell-lab.github.io/cufflinks/cuffcompare/index.html) [18] to compare transcripts and coding genes with those in the refGene library. After predicting the coding functions of the lncRNAs that were identified in this study, we classified the novel lncRNAs into three categories with the most common cluster being “potentially_novel” at 47.32%, followed by “known_coding” at 32.39%, and “undefinable” at 20.29% (Appendix B
Figure A2). After filtering the potentially_novel transcripts, coding potential analysis was performed using CNCI software (https://github.com/www-bioinfo-org/CNCI) [21]. Finally, we obtained 3011 transcripts and 1464 potential lncRNAs, with 93.38% being non-coding, indicating their reliability (Figure 1A).

### 3.3. Features of mRNAs and lncRNAs

To summarize the lncRNAs in our study, we found that 38.56% of the lncRNAs were more than 2000 bp (Figure 1C), whereas 84.02% expanded two or more exons (Figure 1D). All but one of the 1464 lncRNAs were mapped to intronic regions. To ensure high confidence in the lncRNAs identified, we compared them to mRNAs. The preliminary analysis revealed some major differences in gene architecture and expression level among the mRNAs and lncRNAs. For example, the expression levels were higher in mRNAs than lncRNAs (Figure 1B).

### 3.4. Differentially Expressed lncRNAs and mRNAs

The expression levels were analyzed using Cuffdiff software (http://cole-trapnell-lab.github.io/cufflinks/cuffdiff/index.html) [18]. edgeR software (https://bioconductor.org/packages/release/bioc/html/edgeR.html) [24] with parameters of FC ≥ 2.0 and *p*-value ≤ 0.05 was used to identify DELs between the YL and NQ breeds, resulting in 107 DELs, among which 68 DELs were up-regulated and 39 were down-regulated in YL compared to NQ (Figure 2).

### 3.5. Predication of the Targeted Genes

To investigate the function of lncRNAs, we predicted the potential target genes of lncRNAs in *cis* and *trans* actions. For the *cis* action of lncRNAs, we searched for protein-coding genes both 10 and 100 kb upstream and downstream of the lncRNAs. We detected 233 targeted genes, among which *TBX3*, *CACNA1F*, *EDN3*, *KAT5*, *ZNF281*, and *TGFB1* genes are related to body size, which were located near *XLOC_021555*, *XLOC_058541*, *XLOC_048194*, *XLOC_032009*, *XLOC_057255*, and *XLOC_025357*, respectively, thereby suggesting that horse body formation could be regulated by the *cis* actions of several instead of one lncRNA on neighboring protein-coding genes (Table 2). GO analysis of the lncRNA targeted genes demonstrated that most of these genes were clustered into transcription regulations, developmental protein (*p* = 0.01) and positive regulation of mitogen-activated protein kinase (MAPK) activity (*p* = 0.03) pathways. Some genes in the MAPK pathway were also differentially expressed between the inner cell mass and trophectoderm [25]. *TBX3* was clustered into the significantly overrepresented term, developmental protein. These findings demonstrate one of the roles of lncRNAs: the regulation of development (Table 3) through their *cis* actions on their neighboring protein-coding genes, e.g., to regulate the TBX3 protein during body development [26].

### 3.6. Validation of lncRNAs by qRT-PCR

To confirm the expression patterns of the lncRNAs, we performed qRT-PCR analyses on 10 randomly selected lncRNAs identified. Though the theory of RNA-seq and qRT-PCR varies, in RT-PCR, we assumed that the expression of the reference genes is constant for all the samples, whereas in RNA-seq, we assumed that each sample has the same total expressed mRNA. This can be expressed as read count per million mapped reads or RPKM (Reads Per Kilobase per Million mapped reads). When the expression difference exists with sufficient significance, both qRT-PCR and RNA-seq can detect the difference. In this study, qRT-PCR results were aligned with the RNA-seq data, suggesting that the expression patterns based on RNA-seq data were reliable (Figure 3).

## 4. Discussion

In this study, we generated 1.3 billion 150-bp-long paired-end reads of cDNA from six horse placentas. Using the TopHat (http://ccb.jhu.edu/software/tophat/index.shtml) and Cufflinks (http://cole-trapnell-lab.github.io/cufflinks/releases/v2.1.1) programs, 86.3–89.5% of all the reads were successfully mapped against the current reference horse genome EquCab3.0. After developing the bioinformatics pipeline for processing the large amounts of transcriptome sequences, 1464 multiple-exon lncRNAs were identified in the placentas of YL and NQ breeds, and 361 of them overlapped with a previous study [7], which did not contain the placenta tissue, supporting the tissue-specific expression of lncRNAs [27]. We identified 107 lncRNAs as being differentially expressed between the YL and NQ breeds. The most significant phenotypic variation between these two breeds is height. To the best of our knowledge, unique expression profiles of genes associated with specific variations have already been detected in different breeds of chickens [28], pigs [29], and sheep [30].

Among the 233 lncRNA targeted genes, six were related to body development and formation. For instance, the TBX3 (T-box 3) protein is a downstream target of the transforming growth factor-β1 (TGF-β1) signaling pathway and plays an important role in embryonic development [31]. *TBX3* has already been identified as under selection in the Chinese Debao pony [14], suggesting more breeds and samples should be examined to deepen our understanding of the regulation models of horse body size development. *TGF-β1* [25] and *TMED2* (*transmembrane emp24 domain*) were also differentially expressed in embryonic cells and placenta [32], proven by the morphogenesis of the embryo and placenta, as well as skeletal muscle physiology [33]. *CACNA1F* (*L-type voltage-dependent calcium channel a1F*), involved in the differentiation of adipocytes and related to the etiology of obesity [34], was found to be under selection in the Chinese Debao pony [35]. *EDN3* (*endothelin-3*) was reported to be associated with enteric neuron development and blood circulation, thereby influencing the racing performance of Swedish-Norwegian coldblooded trotter horses [36]. *KAT5* (*histone acetyltransferase KAT5*) chromodomain was found to facilitate *SOX4* recruitment to the *CALD1* (*caldesmon 1*) promoter to regulate skeletal myoblast differentiation [37]. As a core transcription factor in embryonic stem cells, *ZNF281* (*zinc finger protein 281*) was found to be related to spontaneous osteochondrogenic differentiation [38]. Other studies showed the strong association of *LCORL/NCAPG*, *HMGA2*, *PROP1*, *LASP* [39], *ZFAT*, *DIAPH3* [40], *ACTN2*, *ADAMTS17*, *GH1*, *ANKRD1* [40], and *ACAN* [41,42] with miniature size and dwarfism in a variety of pony breeds, including Shetland [41,43], miniature [44], Welsh ponies [45], German warmblood horses [46], American miniature horses, Brazilian ponies [47], and Jeju ponies [48], as well as *B4GALT7* [49] and *PROP1* [50] in Friesian horses. These genes were not found in this study, which may be due to the expression specificity of lncRNAs in different horse breeds [27] and different omics levels, since *TBX3*, under strong selection in Chinese ponies, was also identified in this study. This indicated that more breeds and samples are needed to analyze growth regulation during horse development.

We predicted the potential functions of lncRNAs in horse placenta and found that protein-coding genes can interact with lncRNAs through their *cis* actions. In particular, *TBX3*, which has been identified as being associated with horse body size [51], was regulated by two lncRNAs in *cis* actions. We observed that one of these two lncRNAs is located upstream of the enhancer region of the mouse *TBX3* (https://enhancer.lbl.gov/cgi-bin/imagedb3.pl?form=presentation&show=1&experiment_id=483&organism_id=1). As a member of the T-box gene family, *TBX3* plays an important role in embryonic development and some mutations in this gene cause ulnar-mammary syndrome [26]. In the mouse, *TBX3* is recognized as one of the earliest limb initiation factors and loss of functional *TBX3* protein in early and later development may disrupt limb initiation and cause digit loss [52]. The higher expression levels of the two *TBX3*-regulating lncRNAs in YL compared to NQ imply their possible associations with horse height. A certain cluster of lncRNAs in *cis* actions often target protein-coding genes that are specifically expressed in the growth stage (*CACNA1F*, *EDN3*, *KAT5*, *ZNF281*, *TMED2*, and *TGFB1*). These findings suggest the important role of some lncRNAs in horse body development.

Besides the genes mentioned above, more genes accounting for different characteristics were found. *STX17* is a causative gene for melanoma tissue in grey horses [53]. *LGR6* (luteinizing hormone receptor 6) is expressed in embryonic hair follicles, and acts as a marker in the most primitive epidermal stem cells [54]. A mutation in *IQSEC1* can cause intellectual disability, developmental delay, and short stature [55]. *KCNA5* (potassium voltage-gated channel subfamily A member 5) genes, associated with pulmonary arterial hypertension (PAH) [56], may affect the motion of the horse. *COQ2* (coenzyme Q2) and *PLAC8* (placenta-specific 8) are candidate genes for the onset of type 2 diabetes associated with obesity in rats [57].

This study has limitations due to the use of placenta as the main tissue for one Chinese native pony breed versus the Yili horse. Though placenta is not the best tissue for studying the genetic mechanism of animal growth, it is a hotbed for fetal development and using placental tissue is less invasive for the animals, and so it can be used as an alternative tissue for analysis. To reduce interference in breed-specific tissue, we chose the most different characteristic, body size, as the target. In the future, the molecular components underlying the variation in body size in more southwestern Chinese pony breeds should be investigated.

## 5. Conclusions

In this study, we identified the first set of 1464 lncRNAs from horse placentas with 107 of them differentially expressed between the Yili horse and Ningqiang pony breeds. The identification of two specific *TBX3*-regulating lncRNAs suggests their important roles in horse body size. These genetic markers will contribute to the marker-assisted selection of the height of horses.

## Figures and Tables

**Figure 1 animals-10-00119-f001:**
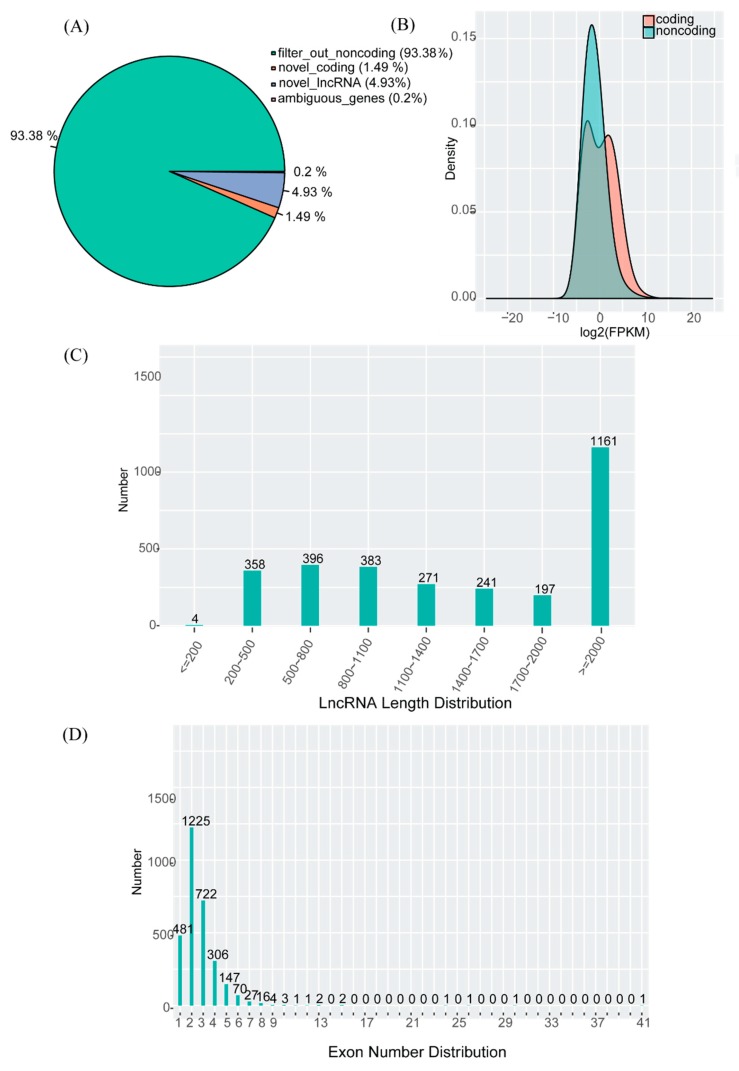
Statistics of horse placental lncRNAs identified in this study. (**A**) Percentage of potentially novel transcripts. (**B**) Expression levels of mRNAs and lncRNAs. (**C**) Length distribution of lncRNAs. (**D**) Exon number of lncRNAs.

**Figure 2 animals-10-00119-f002:**
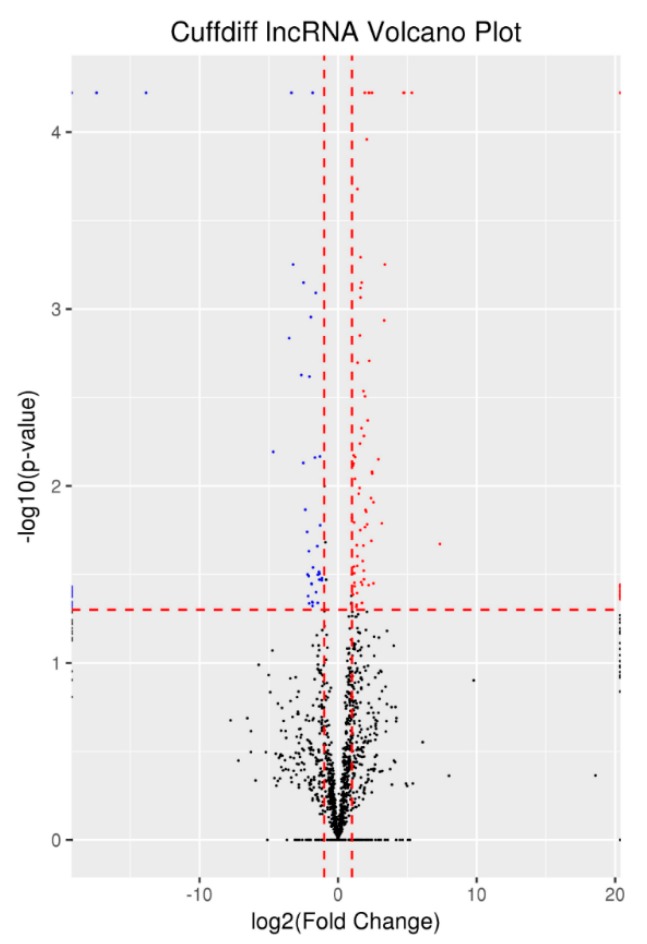
Differentially expressed lncRNAs. Note: The horizonal dashed red line shows criteria of P values equal to 0.05 while vertical dashed lines show log2FC equal to -1 (left) and 1 (right). The points colored with red and blue show the genes if they are upregulated and downregulated, respectively. Black points mean genes does not meet the significant criteria.

**Figure 3 animals-10-00119-f003:**
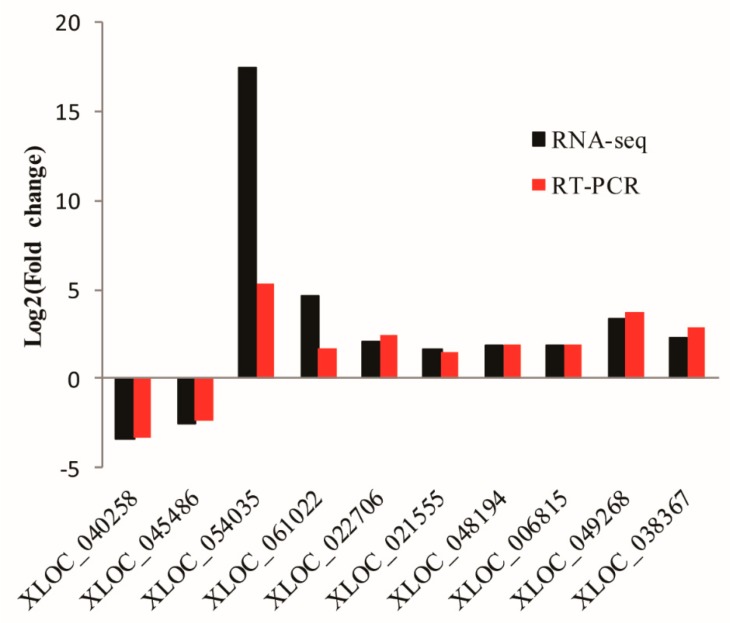
Expression levels of 10 randomly selected lncRNAs based on RNA-seq data and qRT-PCR validation. log2Fold change (FC) > 0 or < 0 indicate the upregulated or down-regulated transcripts in Yili horses compared to Ningqiang ponies, respectively.

**Table 1 animals-10-00119-t001:** Primers for quantitative real-time (qRT)-PCR validation.

lncRNA Name	Length (bp)	Primers	
*XLOC_022706*	204	F	TTACGAGCATCGCTGGGTTT
		R	GGTCTTACAGCGACACCACA
*XLOC_021555*	280	F	GTTCAATGGAGGAGGGCGAA
		R	TGGAGCATCTCGCAAAGTGT
*XLOC_048194*	215	F	AATGCACTTTTCCGGACCCT
		R	CCCCAGTTTGATCGTGTGGA
*XLOC_006815*	217	F	AACGTCGGTCTCTTTTCCCC
		R	AGCAAGTTGTTTCGGGTGGA
*XLOC_049268*	296	F	TGTCGCCCTTCATCTCTGTG
		R	TCAGCGGTGGAAGAAAGCAA
*XLOC_038367*	238	F	TTGCTGGCTGTTGCTTGAAC
		R	TCTCCATTGCCAGTTCGGTG
*XLOC_040258*	178	F	AAACTAACCAGCTCCCCGTG
		R	TTGCCAGCCAAAATGCCTTC
*XLOC_045486*	133	F	CGGAACCTGTTACACTGCCT
		R	CTCACACTTCTGCCCCACAT
*XLOC_054035*	156	F	CGGACAATTACGCAGCCATG
		R	CCTCTGCCCTCCTTAGTCCT
*XLOC_061022*	137	F	CCGTCTTCCTGTTCTGCACT
		R	ACATACACCCTGGCCTGTTG
*β-ACTN* (Beta-actin)	182	F	CAGCCTTCCTTCTTGGGTAT
		R	TGGCATAGAGGTCTTTACGG

Note: F refers to forward primers, R refers to reverse primers.

**Table 2 animals-10-00119-t002:** Significant differentially expressed lncRNAs and their targeted genes.

lncRNAs	Log2FC	*p*	10 kb Upstream	10 kb Downstream	100 kb Upstream	100 kb Downstream
**XLOC_025357**	**1.84**	**5** × 10^−5^			***ERICH4*, *BCKDHA*, *TMEM91*, *DMAC2*, *B3GNT8*, *EXOSC5*, *B9D2*, *TGFB1***	
**XLOC_038367**	**−2.24**	**5** × 10^−5^			***IQSEC1***	
**XLOC_048194**	**1.82**	**5** × 10^−5^			***EDN3***	
**XLOC_051797**	**1.61**	**8** × 10^−4^				***STX17***
XLOC_032225	−3.33	**1.15** × 10^−3^	*PHRF1*	*LMNTD2*, *RASSF7*	*IRF7*, *CDHR5*, *SCT*, *DEAF1*, *PHRF1*, *DRD4*	*LMNTD2*, *HRAS*, *LOC155356*, *RASSF7*, *LRRC56*, *LOC121596*
XLOC_000736	3.53	1.45 × 10^−3^			*PLAU*, *C1H1orf55*, *CAMK2G*	*VCL*
XLOC_050107	2.64	2.35 × 10^−3^				*IFNE*
**XLOC_055900**	**−2.14**	**4.25** × 10^−3^	***IL17REL***		***IL17REL*, *TTLL8***	***ALG12*, *PIM3*, *LOC11177138*, *CRELD2*, *ZBED4***
**XLOC_021555**	**1.66**	**6.9** × 10^−3^			***TBX3***	
XLOC_043830	−2.91	7.05 × 10^−3^	*MFSD1*		*MFSD1*, *LOC15252*	
XLOC_017294	−2.38	0.01			*KIAA1551*	*AMN1*, *ETFBKMT*
XLOC_015361	−2.56	0.01			*TEX44*, *NMUR1*, *LOC1214915*	*PTMA*, *PDE6D*
XLOC_056998	2.36	0.01			*ELF3*, *RNPEP*, *TIMM17A*	*GPR37L1*, *LOC1146338*, *LGR6*, *ARL8A*, *PTPN7*
XLOC_016995	2.23	0.01				*KCNA5*
**XLOC_021660**	**−7.36**	**0.02**	***RILPL1***	***TMED2***	***SNRNP35*, *KMT5A*, *RILPL1*, *RILPL2***	***TMED2*, *DDX55*, *GTF2H3*, *TCTN2*, *ATP6VA2*, *EIF2B1***
XLOC_042377	2.12	0.03			*TSN*, *NIFK*	
**XLOC_059454**	**1.91**	**0.03**			***CD99*, *XG***	***ZBED1*, *DHRSX***
XLOC_008338	−2.20	0.03			*COQ2*, *HPSE*	*LOC163244*, *PLAC8*
**XLOC_022706**	**2.09**	**0.04**			***TBX3***	
**XLOC_027025**	**−1.31**	**0.04**	***LOC17567***	***IGSF23***	***LOC17567*, *CEACAM19*, *CEACAM16***	***ZNF18*, *IGSF23***

Note: log2FC = log2 of fold change. Genes having possible biological functions contributing to growth are indicated in bold font.

**Table 3 animals-10-00119-t003:** Gene Ontology (GO) results of lncRNA targeted genes.

Terms	Counts	*p*	Gene
GO:1903445, protein transport from ciliary membrane to plasma membrane	2	0.018	*RILPL1*, *RILPL2*
GO:0030335, positive regulation of cell migration	5	0.022	*HRAS*, *FOXF1*, *LGR6*, *PLAU*, *TGFB1*
GO:2000630, positive regulation of miRNA metabolic process	2	0.028	*HRAS*, *RELA*
GO:0002513, tolerance induction to self-antigen	2	0.028	*FOXP3*, *TGFB1*
**GO:0043406, positive regulation of MAP kinase activity**	**3**	**0.034**	***EDN3*, *HRAS*, *TGFB1***
**Developmental protein**	**17**	**0.011**	***DEAF1*, *USP7*, *CCM2*, *RNF17*, *ELF3*, *TBX3*, *CAMK2G*, *EN1*, *ACKR3*, *RTL1*, *CITED2*, *CXCL17*, *LBH*, *HAND1*, *HES5*, *FOXC2*, *OLFM1***
Activator	13	0.015	*ATF7IP*, *ELF3*, *RELA*, *ZXDB*, *FOXP3*, *MED13L*, *KAT5*, *ATF7IP2*, *CITED2*, *HAND1*, *IRF7*, *FOXF1*, *FOXC2*
**Transcription regulation**	**31**	**0.025**	***DEAF1*, *ZFP64*, *ZNF18*, *ELF3*, *ASCC1*, *ZXDB*, *ZKSCAN5*, *CITED2*, *LBH*, *HAND1*, *FOXF1*, *OVOL1*, *ZNF394*, *BRD9*, *KMT5A*, *ATF7IP*, *ZNF281*, *FOXL1*, *TBX3*, *RELA*, *GTF2H3*, *FOXP3*, *MED13L*, *KAT5*, *ATF7IP2*, *ZNF789*, *HES5*, *DMRTC2*, *IRF7*, *FOXC2*, *NR5A2***
ecb05205:Proteoglycans in cancer	6	0.029	*HRAS*, *HPSE*, *CAMK2G*, *SDC4*, *PLAU*, *TGFB1*

Note: Clusters associated with growth are indicated in bold font.

## Data Availability

All the Illumina sequencing reads have been deposited in the National Genomics Data Center (https://bigd.big.ac.cn) with the accession code (BioProject ID: PRJCA001875).

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
