# Peer review of "Identification of Novel lncRNAs Differentially Expressed in Placentas of Chinese Ningqiang Pony and Yili Horse Breeds"

_animals, 2020, doi:10.3390/ani10010119_

Round 1

Reviewer 1 Report

The revision of the manuscript by Pu et al, has significantly improved the presentation of the study results. However, I detected some minor issues, which should be solved before publication:

The authors should take care with the term “dwarfism”. I suggest correcting the following sentence by including “miniature size and dwarfism”: Previous studies identified that LCORL/NCAPG, HMGA2, ZFAT [37], 288 and ACAN [38] showed strong association with dwarfism in Shetland, Miniature and Jeju 289 pony breeds… (Line 288-290). Makvandi-Nejad (Four Loci Explain 83% of Size Variation in the Horse) should be additionally cited besides Tetens, as this group was in fact the first, which detected these size-associated genes.

Author Response

The authors should take care with the term “dwarfism”. I suggest correcting the following sentence by including “miniature size and dwarfism”: Previous studies identified that LCORL/NCAPG, HMGA2, ZFAT [37], 288 and ACAN [38] showed strong association with dwarfism in Shetland, Miniature and Jeju 289 pony breeds… (Line 288-290). Makvandi-Nejad (Four Loci Explain 83% of Size Variation in the Horse) should be additionally cited besides Tetens, as this group was in fact the first, which detected these size-associated genes.

Response: We agree that the literature by Makvandi-Nejad is meaningful for horse body size. We have added “miniature size and” in Line 293 and the literature.

Reviewer 2 Report

The revised manuscript does not addressed the flaws in its design.

Author Response

The revised manuscript does not addressed the flaws in its design.

Response: we have added “There are limitations in this study due to the use of placenta as the main tissue and of one Chinese native pony breed versus Yili horse. In the future, it worth to investigate the molecular components underlying the body size variation in more Chinese southwestern pony breeds”.

Round 2

Reviewer 2 Report

In my opinion, no other suggestions for this manuscript.

Author Response

Thankyou for all your suggestions, you helped up to improve our article a lot!

This manuscript is a resubmission of an earlier submission. The following is a list of the peer review reports and author responses from that submission.

Round 1

Reviewer 1 Report

This is a well presented manuscript.  I only had a couple of minor editorial suggestions

line 19 The Ningqiang pony, ....

line 55  , but ..have been... poorly 

line 56 ...only a single study ...has been.....reported in 

line 62 ...but is now close to being endangered -delete nowdays

Author Response

line 19 The Ningqiang pony, ....

Response: We have changed to “The Ningqiang pony” in line 20.

line 55  , but ..have been... poorly 

Response: We have changed it to “ but have been poorly ” in line 57.

line 56 ...only a single study ...has been.....reported in 

Response: We have changed it to “Only a single study has recently reported the identification of horse lncRNAs” in line 58.

line 62 ...but is now close to being endangered -delete nowdays

Response: We have deleted “nowdays” in line 66.

Reviewer 2 Report

Comments to the authors

L19: this first sentence is a fragment, please rephrase

L20, L22, L25 and throughout the paper: try to limit the personal references such as ‘we’, ‘us’, ‘our’ and rather write in an impersonal mode/passive voice

L21: the word ‘second’ is not needed (‘first’ was not stated either)

L29: ‘regulating’ instead of ‘for’; ‘in the horse’ or ‘in horses’

L51: why did you use the past tense saying ‘played’?

L55: ‘in the horse’ or ‘in horses’

L59: please give in brackets all the five Chinese indigenous pony breeds’ names

L61: replace ‘it’ with ‘this breed’

L62: please provide citation

L63: please specify if you refer to artificial or natural selection

L65-66: citation needed

L69: replace ‘for’ with ‘because’ or replace ‘for it acts’ with ‘acting’

L70: delete ‘have’

L75: you cite your own work, so no need to state ‘our’ in the text

L78: remove ‘us’. Rephrase ‘here we performed’ to impersonal mode (e.g. ‘in this study ....... was performed’)

L81: rephrase to impersonal mode

L83: correct ‘may played’

L85: ‘the horse’s body size’ or ‘the horses’ body size’

L92 – L97: Please describe more the selection of the animals for sampling. The mentioned locations are studfarms? You state that the mares were taken from the two locations. Does it mean that these were transported somewhere else? Do the breeders have studbooks for these two breeds? In order to avoid genetic relatedness among the samples random selection of the subjects is not enough, did you study the pedigrees of the selected animals? What do you consider significant genetic relatedness (to be avoided)? If you studied the pedigrees state it. In order to avoid relatedness in which generation the first common ancestor was admitted between the sample mares? If you did not do any pedigree study, please state this. Did you have other registration of the mares to avoid selecting relatives (e.g. sisters, mother and daughter)? Please state which was the closest possible relatedness between the studied animals.

L100: please correct to ‘-80 C°’

L102: you stated in the above subsection that you selected six mares for the study (three YL mares and three NQ mares). Please describe how did you have the nine placenta samples for the study?

L152: you state about the calculation of Pearson’s correlation coefficients, where are the results for these?

L172: replace ‘list’ with ‘listed’

L257: ‘amounts’ instead of ‘amount’

L259: remove ‘to be’

L259: change ‘with previous study’ to ‘with a previous study’

L271-272: please be more specific regarding the study you cite here, as racing performance does not correlate directly just with intrauterine growth of the foal

L291 and L297: offer citations or rephrase, as these statements cannot be drawn based solely on your results (especially regarding other mammalian species except the horse). Otherwise the amplitude of the study in six mares/nine placentas cannot be extrapolated on the whole breed, species.

Author Response

L19: this first sentence is a fragment, ple1ase rephrase

response: We have changed according to your suggestions in line 20-21.

L20, L22, L25 and throughout the paper: try to limit the personal references such as ‘we’, ‘us’, ‘our’ and rather write in an impersonal mode/passive voice

Response: We have changed according to your suggestions.

L21: the word ‘second’ is not needed (‘first’ was not stated either)

response: We have deleted “second” in line 23.

L29: ‘regulating’ instead of ‘for’; ‘in the horse’ or ‘in horses’

Response: we have changed “for” to “regulating”, and “in the horse” to “in horses” in line 29.

L51: why did you use the past tense saying ‘played’?

Response: we have changed “played” to “play” in line 53.

L55: ‘in the horse’ or ‘in horses’

Response: we have changed “in horse” to “in the horse” in line 58.

L59: please give in brackets all the five Chinese indigenous pony breeds’ names

Response: We have added the five Chinese indigenous pony breeds’ names, including Debao, Jianchang, Guizhou, Yunnan and Ningqiang in line 62-63.

L61: replace ‘it’ with ‘this breed’

Response: we have deleted it..

L62: please provide citation

Response: we have added citation: [9]

Animal genetic resources in China:horse, d., camels. China National Commission of Animal Genetic Resources. China Agriculture Press 2011.

L63: please specify if you refer to artificial or natural selection

Response: Ningqiang pony has experienced a long-time selection, both natural and artificial selection, mainly the former. Because it lives in the mountainous area and isolated from outside, but with the development of mechanization, the cultivation of horses is becoming less and less. NQ will more purification in genetics.

L65-66: citation needed

Response: We have added the citation -- Animal genetic resources in China: horse, d., camels. China National Commission of Animal Genetic Resources. China Agriculture Press 2011.

L69: replace ‘for’ with ‘because’ or replace ‘for it acts’ with ‘acting’

Response: we have changed the sentence to “The placenta is essential for fetus growth and development as it acts as a nutrient sensor to regulate the transfer of nutrients from mother to fetus” in line 72-73.

L70: delete ‘have’

Response: we have deleted “have” in line 75.

L75: you cite your own work, so no need to state ‘our’ in the text

Response: We have changed according to your suggestions.

L78: remove ‘us’. Rephrase ‘here we performed’ to impersonal mode (e.g. ‘in this study ....... was performed’)

Response: We have rephrased “In this study, an RNA-seq analysis of the placental tissues from NQ and YL was performed.” line 83.

L81: rephrase to impersonal mode

Response: We have changed according to your suggestions.

L83: correct ‘may played’

Response: we have corrected it to “may play” in line 86.

L85: ‘the horse’s body size’ or ‘the horses’ body size’

Response: we have changed “in horse body” to “horse’s body size” in line 87.

L92 – L97: Please describe more the selection of the animals for sampling. The mentioned locations are studfarms? You state that the mares were taken from the two locations. Does it mean that these were transported somewhere else? Do the breeders have studbooks for these two breeds? In order to avoid genetic relatedness among the samples random selection of the subjects is not enough, did you study the pedigrees of the selected animals? What do you consider significant genetic relatedness (to be avoided)? If you studied the pedigrees state it. In order to avoid relatedness in which generation the first common ancestor was admitted between the sample mares? If you did not do any pedigree study, please state this. Did you have other registration of the mares to avoid selecting relatives (e.g. sisters, mother and daughter)? Please state which was the closest possible relatedness between the studied animals.

Response: The mentioned locations are both National Conservation Farm for the Ningqiang pony and Yili horse breed, respectively. The national farms aim to maintain at least six different pedigrees per one local breed. They keep simple records of pedigree information thus we can require them to collect the samples of different pedigrees. All the placenta tissue samples, kept in RNAlater protection reagent, have been taken from these two farms and transported to our lab by plane in dry ice.

We have added “And to avoid relatedness between the collected samples, we consulted with the local technicians or farmers regarding the pedigrees of the mares. These pregnant mares (multiparous) were randomly selected without any genetic kinship within three generations.” In Line 101-103.

L100: please correct to ‘-80 C°’

Response: we have corrected it in line 105.

L102: you stated in the above subsection that you selected six mares for the study (three YL mares and three NQ mares). Please describe how did you have the nine placenta samples for the study?

Response: Sorry to make this mistake, we took nine samples but sequenced six of them. So, in this article, we used six samples, each three of NQ and YL breeds.

L152: you state about the calculation of Pearson’s correlation coefficients, where are the results for these?

Response: this is for trans prediction, but unluckily, no target genes were predicted.

L172: replace ‘list’ with ‘listed’

Response: we have changed “list” to “listed” in line 179.

L257: ‘amounts’ instead of ‘amount’

Response: we have changed “amount” to “amounts” in line 264.

L259: remove ‘to be’

Response: we have deleted “to be” in line 265.

L259: change ‘with previous study’ to ‘with a previous study’

Response: we have changed “with previous study” to “with a previous study” in line 266.

L271-272: please be more specific regarding the study you cite here, as racing performance does not correlate directly just with intrauterine growth of the foal

Response: we rephrased to “EDN3 (Endothelin-3) was reported to be associated with the enteric neurons development and blood circulation, thereby influencing racing performance of Swedish-Norwegian Coldblooded trotter horses” in line 281-284.

L291 and L297: offer citations or rephrase, as these statements cannot be drawn based solely on your results (especially regarding other mammalian species except the horse). Otherwise the amplitude of the study in six mares/nine placentas cannot be extrapolated on the whole breed, species.

Response: Sorry for this error. We have changed “mammals” to “horse”.

Reviewer 3 Report

In this study, the authors aim to identified the lincRNAs that affect horse body size between the Yili horse and Ningqiang pony breeds by RNA sequencing. The purpose of the research is more meaningful, However, several essential flaws or questions need to addressed and improved. The body type and growth of horses is a complex trait, and the difference in body size between Yili horse and Ningqiang pony is mainly caused by genetic basis rather than the current nutritional and environmental conditions. So, why does the experiment use the maternal placenta as the research subject to screening for lincRNAs affecting body type? Instead of using more suitable fetus as a research material? Therefore, in my opinion, the design of this experiment lacks objectivity and novelty.

Other suggestions:

1、Two horse breeds samples were sequencing, but in ‘3.2 Identification of lincRNAs’ there is only one group result of lincRNA identification, which group of data was used to identify the lincRNA?

2、L208-211 Why compare between mRNA expression levels and lincRNA?

3、L245 The differentially expressed of lincRNA were screened by fold change ≥ 2.0, why define the upregulated or down-regulated by Log2 (fold change) > 0 or < 0 in Figure 3.

4、L248 In the note of table 2, show the ‘Genes having possible biological functions contributing growth are in bold fonts.’, how to define or speculate the genes having possible biological functions, please provide the details in Method.

5、Whether raw data is available online or in some database?

6、A total of 233 target genes of lincRNA were obtained. This paper only discusses the genes that may influence development and growth, and whether it could further speculate on the possible effects on the two breed horse of other target genes between the two groups in Discussion.

Author Response

In this study, the authors aim to identified the lincRNAs that affect horse body size between the Yili horse and Ningqiang pony breeds by RNA sequencing. The purpose of the research is more meaningful, However, several essential flaws or questions need to addressed and improved. The body type and growth of horses is a complex trait, and the difference in body size between Yili horse and Ningqiang pony is mainly caused by genetic basis rather than the current nutritional and environmental conditions. So, why does the experiment use the maternal placenta as the research subject to screening for lincRNAs affecting body type? Instead of using more suitable fetus as a research material? Therefore, in my opinion, the design of this experiment lacks objectivity and novelty.

Response: We agree that fetus will be the most suitable materials for the study of growth. However, due to the limitation of the sample collection we have difficulty to get the fetus for this study. Hopefully in future study we can get the fetus and perform a nicer analysis as the reviewer suggested. In the other hand, the placenta is a primary organ essential for maintaining the proper development of the fetus during pregnancy and it is harmless to the pregnant horses and baby horses when collecting samples.

Other suggestions:

Two horse breeds samples were sequencing, but in ‘3.2 Identification of lincRNAs’ there is only one group result of lincRNA identification, which group of data was used to identify the lincRNA?

Response: We analyze the two groups of data together, and that's all the results.

2、L208-211 Why compare between mRNA expression levels and lincRNA?

Response: because lincRNA expression level is reported lower than that of mRNA, we want to make sure the lincRNA we identified is of high confidence.

3、L245 The differentially expressed of lincRNA were screened by fold change ≥ 2.0, why define the upregulated or down-regulated by Log2 (fold change) > 0 or < 0 in Figure 3.

Response: We used log2 (fold change) to determine whether the expression of lncRNA is up-regulating and down-regulating in sequencing data. While we define it that FC >1 or <-1 were differently expressed in RT-PCR validation. Because, FPKM value was used to detect expression level of lincRNA, while qRT-PCR is tested by comparation of target gene and reference gene [1], it is less comparable and it is impossible to calculate the correlation between these two methods [2]. But we can use the fold of different expression level to test the expression tendency. In this figure, the most meaning what we want to emphasize is that the expression model is similar between RNA-seq and Qrt-PCR, which indicate that the it was validated successfully[3].

4、L248 In the note of table 2, show the ‘Genes having possible biological functions contributing growth are in bold fonts.’, how to define or speculate the genes having possible biological functions, please provide the details in Method.

Response: By querying the literature, we understand the function of the target genes. We have added “The function of target genes was referred to relevant literatures.” In Line 160.

5、Whether raw data is available online or in some database?

Response: we have added “Data availability: All the Illumina sequencing reads have been deposited in the National Genomics Data Center (https://bigd.big.ac.cn) with the accession code (BioProject ID: PRJCA001875).” In Line 310-312.

6、A total of 233 target genes of lincRNA were obtained. This paper only discusses the genes that may influence development and growth, and whether it could further speculate on the possible effects on the two breed horse of other target genes between the two groups in Discussion.

Response: It is true that among the 233 target genes, some will be associated with breed specific other than growth. But the most difference between the two breeds must be the height at withers. So, these genes involved in growth and development can be the candidate genes. The genes chosen were responsible for growth and development of body size, which was consistent with the phenotypic variations in the investigated breeds.

We have added “Though with other different characters, the most significant phenotypic variation between these two breeds is the height at withers. To the best of our knowledge, unique expression profiles of genes associated with specific variations have already been detected in different breeds of chickens [28], pigs [29] and sheep [30].” In Line 269-272.

Reviewer 4 Report

The manuscript by Yabin Pu et al. represents a study on long noncoding RNAs derived from horse placental tissues. It focusses on cis regulatory effects on genes potentially involved in body size determination in Ningqiang pony. This study is interesting and original. I think this approach, using data on lincRNAs from horse tissues is a large step forward into comprehensive research on horse growth. However, what concerns me most is that the literature is not thoroughly reviewed and there are obvious errors in the references section. First of all, the authors should take into account the various studies already done for investigating the genetics of body size in horses in the introduction and discussions section. There have actually been identified main regulators like LCORL, HMGA2, LASP, GH1, OSTN, ADAMTS17 and ZFAT among others. This is an important aspect as ncRNA could affect the expression of these genes. In addition, the study on TBX3 in Debao pony is obviously wrongly citied. I assume [12] must be “Population Variation Reveals Independent Selection toward Small Body Size in Chinese Debao Pony, Kader et al., 2015”. Furthermore, the introductory section on ncRNA and Chinese ponies, although thoroughly written, needs to be verified by references (see particularly line 47-51, 59-68).

In addition to that, I advise the authors to be cautions with their interpretation of the findings. A comparison of Ningqiang and Yili horse is not sufficient to define body size-associated ncRNAs. Yili horses are still of pony size. So, the identified transcriptional differences could be a result of various phenotypic characteristics in-between two breeds but no be related to height at the withers. The authors should admit that larger horses (not ponies) might show different results. This is why I suggest either to do more sequencing on placentas from other horse types (sizes) or to perform a wider data interpretation for these two breed- not only considering size but also other breed specific characteristics.

Furthermore, the actual sampling regions of placental tissues should be defined. In transcriptomics, it is essential for reliable results that the region, sampling time and the way of sampling are the same in cases and controls. This should also be taken into account.

Minor revisions

Simple summary, line 19-20: introductory sentence unclear. The term “height at the withers” should be used among the manuscript.

Line 25: Numbers should be consistently 3011 or 3,011

Line 29: …body size in horses as well as…

Abstract, line 34: at this stage the reader already need the information that Yili horse is larger than Nigqiang pony

Introduction, line51: The intergenic lincRNAs play an important…

Line 55: …identified in horses and other animals, but poorly characterized in the horse. Only…

Line 56-58: This sentence is about results- no introduction

Line 63: “hardy body” term unclear

Line 70: Previous studies have shown…

Line 95: The actual height at the withers (average/mean) should be given for the sampled individuals.

Line 117: “of the clean data” is given double times

Figure A3 is missing! (see line 197 and appendix A)

Author Response

The manuscript by Yabin Pu et al. represents a study on long noncoding RNAs derived from horse placental tissues. It focusses on cis regulatory effects on genes potentially involved in body size determination in Ningqiang pony. This study is interesting and original. I think this approach, using data on lincRNAs from horse tissues is a large step forward into comprehensive research on horse growth. However, what concerns me most is that the literature is not thoroughly reviewed and there are obvious errors in the references section.

Response: Sorry to make these errors. We have revised all through the article, as well as the reference part. we also asked senior specialist to helped us revise it.

First of all, the authors should take into account the various studies already done for investigating the genetics of body size in horses in the introduction and discussions section. There have actually been identified main regulators like LCORL, HMGA2, LASP, GH1, OSTN, ADAMTS17 and ZFAT among others. This is an important aspect as ncRNA could affect the expression of these genes.

Response: This is a good suggestion to improve article. Many genes had been found be associated with dwarfism of pony breeds. It is meaningful to connect these genes and expression models. We have added “Previous studies identified that LCORL/NCAPG, HMGA2, ZFAT [40], and ACAN [41] showed strong association with dwarfism in Shetland, Miniature and Jeju pony breeds. Unfortunately, these genes were not found in this study. It may be due to the expression specificity of lincRNAs in different horse breeds [42], since the TBX3 gene, under strong selection in Chinese Pony, was also identified in this study. This indicated that more breeds and samples are needed to analyze the regulation during horse development.” In Line 288-293.

In addition, the study on TBX3 in Debao pony is obviously wrongly citied. I assume [12] must be “Population Variation Reveals Independent Selection toward Small Body Size in Chinese Debao Pony, Kader et al., 2015”. Furthermore, the introductory section on ncRNA and Chinese ponies, although thoroughly written, needs to be verified by references (see particularly line 47-51, 59-68).

Response: Sorry to bring misunderstanding for you. In this article, we found that gene TBX3 is under strong selection in Chinese Debao pony. We have added references to lncRNA and ponies in the first and second paragraph.

In addition to that, I advise the authors to be cautions with their interpretation of the findings. A comparison of Ningqiang and Yili horse is not sufficient to define body size-associated ncRNAs. Yili horses are still of pony size. So, the identified transcriptional differences could be a result of various phenotypic characteristics in-between two breeds but no be related to height at the withers. The authors should admit that larger horses (not ponies) might show different results. This is why I suggest either to do more sequencing on placentas from other horse types (sizes) or to perform a wider data interpretation for these two breed- not only considering size but also other breed specific characteristics.

Response: The official definition of a pony is a horse that measures less than 14.2 hands (58 inches, 147 cm) at the withers (Mattern 2010). In China, horses with height at withers less than 106 cm are defined as pony. Thus, NQ can be defined as pony. YL horse with height about 148 cm is significantly taller than Ningqiang Ponies. This huge variation is the major difference causing the differential expression of the lincRNA genes.

Mattern J. 2010. Horses on the farm. Rourke Pub Group.

Furthermore, the actual sampling regions of placental tissues should be defined. In transcriptomics, it is essential for reliable results that the region, sampling time and the way of sampling are the same in cases and controls. This should also be taken into account.

Response: We have revised the section of Animals and samples in the methods part as following:

The placental samples of Ningqiang pony (NQ) and Yili horse (YL) were taken from Ningqiang national conservation farm in Ningqiang County of Shaanxi Province (log: 105.582 and lat: 32.825) and Zhaosu horse farm in Zhaosu County of Xinjiang Province (log: 81.131 and lat: 43.155), respectively. Three YL mares with an average adult height of 141.3 cm and three NQ mares with adult an average height less than 106 cm were taken from these two areas. To make sure the consistency between biological replications, adult mares about six years old were chosen for placental tissue sampling. And to avoid relatedness between the collected samples, we consulted with the local technicians or farmers regarding their pedigrees of the mares. These pregnant mareshorses (multiparous) were randomly selected without any genetic kinship within three generations to avoid genetic relatedness among the samples. After giving births, the placental tissues fromof the mares were immediately collected by experienced veterinarians and well stored in the RNAlater (Thermo Fisher Scientific, USA) at -80 oC for RNA extraction. All the samples were transported to our laboratory ibrary for RNA extraction and later sequencing.

Minor revisions

Simple summary, line 19-20: introductory sentence unclear. The term “height at the withers” should be used among the manuscript.

Response: we have changed to “Ningqiang pony (NQ) is a famous breed with an average adult height at withers less than 106 cm, but the genetic mechanism responsible for its small stature remains unclear.” in line 20-21.

Line 25: Numbers should be consistently 3011 or 3,011

Response: We have changed to 3,011 in Line 38.

Line 29: …body size in horses as well as…

Response: we have changed it to “This finding suggests the potential functional significance of placental lncRNAs in regulating horse body size.” In Line 45-46.

Abstract, line 34: at this stage the reader already need the information that Yili horse is larger than Nigqiang pony

Response: We have added “NQ, a kind of pony breeds with adult height at withers less than 106 cm, while much shorter than YL horse (148 cm).” in Line 35-36.

Introduction, line51: The intergenic lincRNAs play an important…

Response: we have changed “played” to “play” in line 53.

Line 55: …identified in horses and other animals, but poorly characterized in the horse. Only…

Response: we have changed it to “Large numbers of lncRNAs have been identified in a wide spectrum of organisms[4-6], but have been poorly characterized in horse [7].” In Line 56-57.

Line 56-58: This sentence is about results- no introduction

Response: we rephrased to “Only a single study has recently reported the identification of horse lncRNAs” In Line 58.

Line 63: “hardy body” term unclear

Response: we changed it to “tough body” in line 66.

Line 70: Previous studies have shown…

Response: we have changed it in line 70.

Line 95: The actual height at the withers (average/mean) should be given for the sampled individuals.

Response: It has been given in Line 98.

Line 117: “of the clean data” is given double times

Response: we have deleted it.

Figure A3 is missing! (see line 325 and appendix A)